# Phytochemical Composition and Content of Red-Fleshed Grape Accessions

Lizhen Lu [1,2,3], Yingzhen Yang [4], Gan-Yuan Zhong [4], Zhenchang Liang [1,2,*] and Lailiang Cheng [5,*]

1   Beijing Key Laboratory of Grape Science and Wine Technology, and Key Laboratory of Plant Resources, Institute of Botany, Chinese Academy of Sciences, Beijing 100093, China; lulizhen@ibcas.ac.cn
2   China National Botanical Garden, Beijing 100093, China
3   University of Chinese Academy of Sciences, Beijing 100049, China
4   USDA-ARS Grape Genetics Research Unit, Geneva, NY 14456, USA
5   Section of Horticulture, School of Integrative Plant Science, Cornell University, Ithaca, NY 14853, USA
*   Correspondence: zl249@ibcas.ac.cn (Z.L.); lc89@cornell.edu (L.C.); Tel.: +86-010-62836079 (Z.L.); +1-607-255-1779 (L.C.); Fax: +86-010-62836079 (Z.L.); +1-607-255-0599 (L.C.)

**Abstract:** Red-fleshed grapes are important breeding resources, and study of the content and composition of phenolic compounds in red-fleshed grapes is lacking. In this study, the profiles of phenolic compounds in the whole berry, flesh, and peel of thirteen red-fleshed grape (*Vitis*) accessions were determined for two consecutive years. The content of total phenolic compounds ranged from 4.795 to 29.875 mg g$^{-1}$ FW (fresh weight) in berry, from 1.960 to 12.593 mg g$^{-1}$ FW in flesh, and from 17.067 to 60.182 mg g$^{-1}$ FW in peel. As expected, anthocyanins were the main phenolic compounds, accounting for 90.4, 89.4, and 94.1% of the total phenolic compounds in berry, flesh, and peel, respectively. Flavanols accounted for 36.2% of the non-anthocyanin phenolic compounds in berry, 35.3% in flesh, and 38.3% in peel. In comparison, flavonols accounted for about 11.6, 5.7, and 15.8% of the non-anthocyanin phenolic compounds in berry, flesh, and peel, respectively. Hydroxycinnamic derivatives were the most abundant non-anthocyanins and accounted for 53.8, 56.1, and 44.3% of non-anthocyanin phenolic compounds in these three tissues. The content of phenolic compounds in peel was significantly higher than that in flesh and whole berry. A significant variation in the content of total and individual phenolic compounds was observed among different red-fleshed grapes, suggesting that genetic background was an important factor affecting the accumulation of these phenolic compounds. This work represents the most comprehensive characterization of phenolic compounds profiles in red-fleshed grapes.

**Keywords:** red-fleshed grape; phenolic compounds; anthocyanins; flavanols; teinturier





## 1. Introduction

Phenolic compounds are one class of important secondary metabolites in grape berry, including anthocyanins, flavan-3-ols, flavonols, and phenolic acids [1–3]. Among these groups of phenolic compounds, anthocyanins are the most abundant one in red grapes, which impact the pigmentation of grapes, wines, and other processed products [4–10]. Flavan-3-ols are the building blocks of grape tannins and have a direct impact on the complexity of wine taste and mouthfeel [11–14]. Flavonols, as glycosides, are important co-factors for the color enhancement of wine [14–16]. Phenolic acids play critical roles in developing the bitterness and astringency properties of wine and human nutrition [17–20]. In recent years, phenolic compounds have attracted much attention as antioxidant and bioactive compounds because of their beneficial effects on human health [21–28].

There is a wide variation in the composition and content of anthocyanins and other groups of phenolic compounds across different types of grapes [29–32]. Because anthocyanins are the most abundant type of phenolic compounds and they are primarily present in colored tissue, red grapes usually have a higher level of anthocyanins content. The total

phenolic compounds content of red grape is higher than that in white skin grapes [31–34]. Most grapes have non-colored berry flesh, but there are a small number of grape cultivars with red berry flesh. Well-known red-flesh grape cultivars include 'Alicante Bouschet', 'Carmina', 'Deckrot', 'Gamay Teinturier', 'Grand Noir de la Calmette', 'Kolor', 'Pinot Teinturier', 'Royalty', 'Rubired', 'Salvador', 'Saperavi', 'Siebouschet', 'Sulmer', 'Petit Bouschet', and 'Rubired' [35–37]. Red-flesh grapes, as expected, have higher levels of anthocyanins than most non-red-flesh cultivars. Castillo-Munoz et al. [38] found that the anthocyanins content in the whole berry of 'Garnacha Tintorera' ('Alicante Bouschet') was 2441 and 3593 mg kg$^{-1}$ in two different vineyards, respectively. Similarly high levels of anthocyanins content were observed in several other studies of red-flesh grapes [39–44]. He et al. [45] documented different composition and content of anthocyanins in skin and flesh tissues in a Chinese red wine cultivar 'Yan 73'. Liang et al. [29] analyzed the composition and content of anthocyanins and several other types of polyphenolic compounds in 344 *V. vinifera* cultivars including two red-flesh grape cultivars, 'Salvador' and 'Royalty', and found that the berry anthocyanin content in the red-flesh grapes of 'Salvador' and 'Royalty' were 3.7 and 5.1 times higher, respectively, than the mean anthocynin content of *V. vinifera* with colored skin. Similarly, the content of flavanols, flavonols, hydroxycinnamic derivatives, and hydroxybenzoic acids in 'Salvador' and 'Royalty' were all higher than the mean content of the corresponding compounds in the non-red-flesh *V. vinifera* cultivars [29–32,46].

Because red-fleshed grapes have high levels of anthocyanins and other phenolic compounds which are health-beneficial, there is an increasing interest in breeding red-flesh grape varieties for the juice and table grape industry [34,47,48]. While many existing red-flesh grapes may originate from similar sources [37], manifestation of the red flesh trait and therefore the content of anthocyanins are expected to vary with genetic background. Previous studies of red-flesh grapes were limited to one or two varieties and mainly focused on anthocyanins [43,49,50]. In this study, we analyzed the composition and content of all the major phenolic compounds, including anthocyanins, flavanols, flavonols, hydroxycinnamic derivatives, and hydroxybenzoic acids in the whole berry (excluding seeds), skin, and flesh in 13 red-flesh grape accessions preserved in the USDA-Agricultural Research Service *Vitis* Clonal Repository in Geneva, New York (Table 1). These 13 red-flesh grapes have very diverse and complex pedigrees, which offer an excellent opportunity for evaluation of the manifestation of the red flesh trait in different genetic backgrounds. This study is a comprehensive characterization of the phenolic compounds in the red-flesh *Vitis* germplasm preserved in the repository, and the results will have important value for the breeding of red-fleshed grape cultivars in the future.

**Table 1.** Red-fleshed grape accessions used in this study.

| No | Cultivar Name | Accession ID (PI No.) | Pedigree | Origin |
|---|---|---|---|---|
| 1 | Sori | 588103 [a] | *V. acerifolia* x *V. riparia* | Unknown |
| 2 | Pulliat | 588190 | Herbemont seedling, bourq., *vinifera* (all according to Loomis card), note Loomis does not show *riparia* for Pulliat | United States |
| 3 | Bailey Alicante | 588361 | Bailey Alicante | Unknown |
| 4 | NY 65.548.3 | 588521 | Ill 791-1 (Jaeger 70 x Victoria's Choice) x Ill 820-1 (Ill 271-1 x Black Monukka) | United States |
| 5 | Bailey Alicante A | 588579 | Bailey x Alicante Bouschet | Japan |
| 6 | Agria | 588670 | Unknown | Hungary |
| 7 | GVIT 1392 | 588681 | *V. rupestris* x *V. vulpina* | United States |
| 8 | Landot 234 | 597158 | Seibel 5455 x Gamay Mourot | Unknown |
| 9 | Seibel 6339 | 597175 | Seibel 867 x Seibel 2524–*V. cinerea, V. labrusca, V. lincecumii, V. riparia, V. rupestris, V. vinifera* | France |
| 10 | Seibel 4646 | 597193 | Seibel 2508 x Seibel 880–*V. aestivalis, V. cinerea, V. rupestris, V. vinifera* | France |
| 11 | HN 12 | 597259 | Seibel 10878 x Couderc 299-35 | Unknown |
| 12 | Rubaiyat | 597289 | Seibel 5437 x Bailey | United States |
| 13 | Seibel 5437 | GVIT1616 | Seibel 867 x Seibel 2512 | France |

[a]: All accessions information can be found in USDA-ARS Germplasm Resources Information Network "https://www.ars-grin.gov/ (accessed on 10 January 2020)".

## 2. Materials and Methods

### 2.1. Plant Material

Berry samples of 13 red-flesh grape accessions were harvested upon their ripening from the USDA-ARS *Vitis* Clonal Repository in Geneva, New York in two consecutive years (2009 and 2010, Table 1). Grape berries were randomly sampled at ripening, which was determined upon seed color change to dark brown without senescence and measurements of total soluble solid (Brix) of berries. Brix data were collected using a hand-held Atago PR-32α Palette digital refractometer. All the vines received standard fertilization, irrigation, pruning, and insect and disease control.

There were two grapevines for each accession. About 100 representative berries were collected from each individual vine. The numbers of berries were counted, and the berry weight was recorded for each sample before being frozen and stored at −80 °C for further processing.

### 2.2. Extraction of Phenolic Compounds

The extraction method of phenolic compounds was the same as that described by Liang et al. [29] with minor modification. Briefly, the sample was divided into two halves. For the half to be used for measuring phenolic compounds in the whole berry, the frozen berries were crashed using a mortar and pestle. After removing all the seeds, flesh, and peel, tissues were ground in an IKA A11 mill (IKA Works Inc., Wilmington, NC, USA) while frozen. The other half was manually peeled, all the seeds were removed, and the resulting skins and flesh were separately frozen in liquid $N_2$ immediately, and then ground in an IKA A11 mill. For all the tissue types, about 0.5 g powdery sample was weighed for analysis separately. The powdery samples were ground in 5 mL extraction solution (2:28:70, formic acid/water/methanol) using mortar and pestle. The extracts were shaken in a thermomixer (Eppendorf, Hamburg, HH, Germany) for 10 min. Then, the extracts were centrifuged at 13,000× *g* at 4 °C for 10 min. About 1 mL extract was filtered through a 0.2 μm membrane filter (Agilent Technologies, Santa Clara, CA, USA) for analysis.

### 2.3. Qualitative and Quantitative Analysis of Phenolic Compounds

We followed the same analysis methods of Liang et al. [29] for identifying and quantifying phenolic compounds. High-performance liquid chromatography/quadrupole-time of flight mass spectrometer (HPLC/Q-TOF MS/MS) (Micromass Q-TOF micro, Waters, Milford, MA, USA) was employed for identifying phenolic compounds. The system was equipped with a Waters Alliance 2695 HPLC Pump, Waters Alliance 2695 Autosampler and Waters 996 photodiode array detector, which were coupled directly to the sprayer needle where ions were generated by electrospay ionization (ESI) in both positive and negative ionization modes. A reverse-phase C18 Inertsil ODS-3 column (5 μm particle sizes, 250 mm × 4.6 mm I.D.) from GL Sciences (Tokyo, Honshu, Japan) and a C18 Nova Pack guard column (Waters, Milford, MA, USA) were used for the analysis. The mobile phase consisted of water/formic acid (90:10) as solvent A, and acetonitrile/formic acid (90:10) as solvent B. The gradient profile began at 95% A, to 85% A at 25 min, 73% A at 53 min, 95% A at 57 min, and remained at 95% A for 5 min. The flow rate was 1.0 mL min$^{-1}$ and the column temperature was set at 30 °C. The injection volume was 20 μL. Phenolic compounds were detected at 280, 320, 360, and 520 nm on the diode array detector, and at the same time, spectrum scans were performed from 210 nm to 600 nm. For MS analyses, nitrogen was used as the drying and nebulizing gas, and the nebulizer pressure was 380 Pa. Gas flow was set at 10 L min$^{-1}$ and temperature was 350 °C. The capillary voltage was 3000 V. Mass spectra of anthocyanins and other polyphenolic compounds were recorded in both positive and negative ionization modes between *m/z* 100 and 1000, respectively.

The same HPLC protocol was used in the Agilent 1100 HPLC system (Aglient Corporation, Palo Alto, CA, USA) fitted with an Agilent 1100 diode array detector and an autosampler for quantifying phenolic compounds for all samples. The concentration of individual phenolic compounds was quantified based on peak area and standard curves derived

from corresponding authentic polyphenolic compounds as described in Liang et al. [30]. All phenolic compounds standards were obtained from Sigma-Aldrich (St. Louis, MO, USA), Extrasynthese (Lyon, Rhone, France), and AApin Chemicals (Abingdon, Oxon, UK).

### 2.4. Statistical Analyses

Data analysis was carried out using the SAS 9.2 package (SAS Institute Inc., Cary, NC, USA). SAS programs of GLM were used for mean separation and significance testing ($p < 0.05$). The boxplot was developed by using Sigmaplot 10.0 for Windows (SPSS, Chicago, IL, USA). Each accession only had two plants, so analysis of variance was carried out by treating data from 2009 and 2010 data as replicates, and we had a total of 4 replicates.

## 3. Results and Discussion

### 3.1. Identification of Phenolic Compounds

On the basis of the retention time, peak area, molecular ions, important fragment ions, and UV-Vis spectra absorbance maxima generated from MS and HPLC profiles, 48 phenolic compounds were detected in 13 red-fleshed grape accessions (Tables 1 and 2). These compounds can be classified into five groups according to their chemical structure: 28 anthocyanins, 2 hydroxybenzoic acids, 6 hydroxycinnamic derivatives, 6 flavonols, and 6 flavanols. The anthocyanins detected in this study mainly consisted of mono- and di-glucoside derivatives of 5 anthocyanidins: delphinidin (Dp), cyanidin (Cy), petunidin (Pt), peonidin (Pn), and malvidin (Mv). In addition, their acylation derivatives (6-*O*-acetyl, 6-*O*-coumaryl and 6-*O*-caffeoyl) were also detected.

**Table 2.** Phenolic compounds identified by chromatography and mass spectrometry and their mean values of 13 red-flesh *Vitis* accessions in whole berry, flesh, and peel (mg g$^{-1}$ FW, Mean ± SE).

| No | RT (min) | Molecular Ion M (*m/z*) | Fragment Ions M (*m/z*) | Absorbance Maxima (nm) | Identity | Berry (mg g$^{-1}$ FW) | Flesh (mg g$^{-1}$ FW) | Peel (mg g$^{-1}$ FW) |
|---|---|---|---|---|---|---|---|---|
| 1 | 4.13 | 170 | 170 | 280 | Gallic acid | 0.006 ± 0.001 b * | 0.003 ± 0.001 c | 0.011 ± 0.002 a |
| 2 | 15.94 | 168 | 168 | 280 | Vanillic acid | 0.013 ± 0.002 b | 0.015 ± 0.003 b | 0.033 ± 0.008 a |
| 3 | 6.95 | 578 | 577 | 280 | **Procyanidin B1** | 0.111 ± 0.013 b | 0.061 ± 0.006 c | 0.147 ± 0.016 a |
| 4 | 9.63 | 290 | 290 | 276 | **Catechin** | 0.121 ± 0.016 b | 0.068 ± 0.010 c | 0.207 ± 0.026 a |
| 5 | 16.34 | 290 | 289 | 280 | Epicatechin | 0.044 ± 0.005 b | 0.026 ± 0.004 c | 0.084 ± 0.011 a |
| 6 | 23.3 | 442 | 290 | 280 | Epicatechin gallate | 0.036 ± 0.004 b | 0.018 ± 0.002 c | 0.067 ± 0.011 a |
| 7 | 35.96 | 316 | 315 | 280 | Isorhamnetin | 0.020 ± 0.001 b | 0.008 ± 0.001 c | 0.08 ± 0.012 a |
| 8 | 39.59 | 161 | 161 | 280 | Tryptophol | 0.010 ± 0.001 b | 0.004 ± 0.000 c | 0.025 ± 0.001 a |
| 9 | 10.34 | 578 | 577 | 280 | Procyanidin B2 | 0.092 ± 0.008 b | 0.033 ± 0.006 c | 0.307 ± 0.036 a |
| 10 | 7.9 | 312 | 180 | 318 | **Caftaric acid** | 0.461 ± 0.025 a | 0.254 ± 0.020 b | 0.518 ± 0.073 a |
| 11 | 11.56 | 296 | 164 | 331 | **Coutaric acid** | 0.172 ± 0.011 b | 0.057 ± 0.012 c | 0.418 ± 0.068 a |
| 12 | 11.83 | 354 | 354 | 320 | Chlorogenic acid | 0.020 ± 0.003 a | 0.017 ± 0.002 a | 0.029 ± 0.003 a |
| 13 | 26.98 | 194 | 194 | 320 | Ferulic acid | 0.023 ± 0.001 b | 0.006 ± 0.001 c | 0.032 ± 0.004 a |
| 14 | 41.76 | 228 | 228 | 318 | Resveratrol | 0.003 ± 0.000 b | 0.001 ± 0.000 c | 0.013 ± 0.002 a |
| 15 | 38.34 | 449 | 287 | 320 | Kaempferol 3-*O*-glucoside | 0.005 ± 0.004 b | 0.002 ± 0.001 c | 0.011 ± 0.007 a |
| 16 | 23.6 | 480 | 318 | 365 | Myricetin 3-*O*-glucoside | 0.025 ± 0.001 b | 0.009 ± 0.000 c | 0.048 ± 0.002 a |
| 17 | 29.42 | 610 | 609 | 365 | **Rutin** | 0.062 ± 0.002 b | 0.002 ± 0.000 c | 0.134 ± 0.029 a |
| 18 | 30.65 | 478 | 302 | 356 | Quercetin 3-*O*-glucuronide | 0.008 ± 0.001 b | 0.002 ± 0.000 c | 0.051 ± 0.006 a |
| 19 | 31.78 | 464 | 302 | 355 | Quercetin 3-*O*-glucoside | 0.032 ± 0.003 b | 0.009 ± 0.001 c | 0.049 ± 0.007 a |
| 20 | 39.95 | 479 | 317 | 350 | Isorhamnetin 3-*O*-glucoside | 0.017 ± 0.001 b | 0.010 ± 0.001 c | 0.075 ± 0.009 a |
| 21 | 9.51 | 627 | 303, 465 | 282, 520 | Delphinidin 3-*O*-glucoside-5-*O*-glucoside | 0.730 ± 0.031 b | 0.284 ± 0.045 c | 2.911 ± 0.325 a |
| 22 | 12.64 | 611 | 287, 449 | 282, 516 | Cyanidin 3-*O*-glucoside-5-*O*-glucoside | 0.15 ± 0.009 b | 0.083 ± 0.011 c | 0.492 ± 0.081 a |
| 23 | 14.22 | 465 | 303 | 280, 523 | **Delphinidin 3-*O*-glucoside** | 2.299 ± 0.098 b | 0.876 ± 0.072 c | 7.26 ± 0.451 a |
| 24 | 14.86 | 641 | 317, 479 | 274, 523 | Petunidin 3-*O*-glucoside-5-*O*-glucoside | 0.781 ± 0.036 b | 0.286 ± 0.030 c | 2.29 ± 0.201 a |
| 25 | 17.85 | 449 | 287 | 279, 515 | Cyanidin 3-*O*-glucoside | 0.325 ± 0.025 b | 0.218 ± 0.023 b | 0.655 ± 0.064 a |
| 26 | 18.53 | 625 | 301, 463 | 278, 513 | Peonidin 3-*O*-glucoside-5-*O*-glucoside | 0.58 ± 0.025 b | 0.471 ± 0.068 b | 1.292 ± 0.159 a |
| 27 | 20.31 | 479 | 317 | 277, 526 | **Petunidin 3-*O*-glucoside** | 1.614 ± 0.060 b | 0.849 ± 0.048 c | 4.216 ± 0.341 a |
| 28 | 20.53 | 655 | 331, 493 | 275, 524 | Malvidin 3-*O*-glucoside-5-*O*-glucoside | 1.474 ± 0.069 b | 0.452 ± 0.052 c | 4.358 ± 0.257 a |
| 29 | 23.82 | 653 | 287, 449, 611 | 280, 516 | Cyanidin 3-*O*-(6-*O*-acetyl)-glucoside-5-*O*-glucoside | 0.003 ± 0.000 a | 0 ± 0.000 b | 0.005 ± 0.000 a |
| 30 | 24.58 | 463 | 301 | 279, 515 | Peonidin 3-*O*-glucoside | 0.445 ± 0.027 b | 0.331 ± 0.038 b | 0.859 ± 0.084 a |
| 31 | 25.83 | 683 | 317, 479, 641 | 280, 530 | Petunidin 3-*O*-(6-*O*-acetyl)-glucoside-5-*O*-glucoside | 0.013 ± 0.001 b | 0.004 ± 0.001 c | 0.038 ± 0.007 a |
| 32 | 27.05 | 493 | 331 | 278, 530 | **Malvidin 3-*O*-glucoside** | 1.571 ± 0.055 b | 0.65 ± 0.057 c | 4.564 ± 0.253 a |
| 33 | 27.79 | 773 | 287, 611 | 281, 525 | Cyanidin 3-*O*-(6-*O*-caffeoyl)-glucoside-5-*O*-glucoside | 0.002 ± 0.000 b | 0.001 ± 0.000 b | 0.006 ± 0.001 a |

**Table 2.** *Cont.*

| No | RT (min) | Molecular Ion M ($m/z$) | Fragment Ions M ($m/z$) | Absorbance Maxima (nm) | Identity | Berry (mg g$^{-1}$ FW) | Flesh (mg g$^{-1}$ FW) | Peel (mg g$^{-1}$ FW) |
|---|---|---|---|---|---|---|---|---|
| 34 | 29.75 | 507 | 303, 465 | 280, 521 | Delphinidin 3-*O*-(6-*O*-acetyl)-glucoside | 0.078 ± 0.003 b | 0.018 ± 0.002 c | 0.263 ± 0.034 a |
| 35 | 33.42 | 491 | 287 | 279, 514 | Cyanidin 3-*O*-(6-*O*-acetyl)-glucoside | 0.014 ± 0.001 b | 0.007 ± 0.002 c | 0.084 ± 0.011 a |
| 36 | 34.37 | 773 | 303, 465, 627 | 279, 530 | Delphinidin 3-*O*-(6-*O*-coumaryl)-glucoside-5-*O*-glucoside | 0.133 ± 0.009 b | 0.049 ± 0.009 b | 0.565 ± 0.027 a |
| 37 | 36.5 | 521 | 317, 479 | 280, 530 | Petunidin 3-*O*-(6-*O*-acetyl)-glucoside | 0.045 ± 0.002 b | 0.007 ± 0.001 c | 0.229 ± 0.017 a |
| 38 | 38.2 | 757 | 287, 449, 611 | 280, 524 | Cyanidin 3-*O*-(6-*O*-coumaryl)-glucoside-5-*O*-glucoside | 0.013 ± 0.001 b | 0.008 ± 0.001 c | 0.083 ± 0.010 a |
| 39 | 39.74 | 505 | 301 | 280, 518 | Peonidin 3-*O*-(6-*O*-acetyl)-glucoside | 0.062 ± 0.003 b | 0.014 ± 0.002 c | 0.205 ± 0.025 a |
| 40 | 41.12 | 773 | 303, 465, 627 | 282, 530 | Delphinidin 3-*O*-(6-*O*-coumaryl)-glucoside | 0.305 ± 0.025 b | 0.065 ± 0.007 c | 1.005 ± 0.084 a |
| 41 | 42.4 | 535 | 331, 493 | 280, 521 | Malvidin 3-*O*-(6-*O*-acetyl)-glucoside | 0.203 ± 0.016 b | 0.044 ± 0.007 c | 0.894 ± 0.067 a |
| 42 | 43.38 | 783 | 317, 479, 641 | 280, 530 | Petunidin 3-*O*-(6-*O*-coumaryl)-glucoside-5-*O*-glucoside | 0.039 ± 0.004 b | 0.016 ± 0.003 c | 0.143 ± 0.021 a |
| 43 | 43.95 | 771 | 301, 463, 625 | 279, 520 | Peonidin 3-*O*-(6-*O*-coumaryl)-glucoside-5-*O*-glucoside | 0.061 ± 0.002 b | 0.02 ± 0.004 c | 0.232 ± 0.046 a |
| 44 | 45.23 | 801 | 331, 493,655 | 280, 530 | Malvidin 3-*O*-(6-*O*-coumaryl)-glucoside-5-*O*-glucoside | 0.132 ± 0.008 b | 0.052 ± 0.008 c | 0.654 ± 0.088 a |
| 45 | 46.98 | 595 | 287, 449 | 283, 522 | Cyanidin 3-*O*-(6-*O*-coumaryl)-glucoside | 0.194 ± 0.012 b | 0.047 ± 0.004 c | 0.553 ± 0.063 a |
| 46 | 48.78 | 625 | 317, 479 | 280, 531 | Petunidin 3-*O*-(6-*O*-coumaryl)-glucoside | 0.203 ± 0.012 b | 0.074 ± 0.007 c | 1.140 ± 0.131 a |
| 47 | 51.35 | 609 | 301, 463 | 279, 523 | Peonidin 3-*O*-(6-*O*-coumaryl)-glucoside | 0.078 ± 0.008 b | 0.031 ± 0.002 c | 0.224 ± 0.026 a |
| 48 | 52.59 | 639 | 331, 493 | 280, 521 | Malvidin 3-*O*-(6-*O*-coumaryl)-glucoside | 0.510 ± 0.032 b | 0.148 ± 0.014 c | 1.705 ± 0.156 a |

\*: The bold are the main phenolic compounds, and those with different letters in the same rows are significantly different at $p < 0.05$.

Large variation was detected for 48 phenolic compounds in 13 red-fleshed grape accessions (Table 2). Anthocyanins were the major phenolic compounds in berry, flesh, and peel. Delphinidin 3-*O*-glucoside was the most abundant compound in the group of anthocyanins (2.299, 0.876, and 7.260 mg g$^{-1}$ FW in berry, flesh, and peel, respectively), followed by petunidin 3-*O*-glucoside (1.614, 0.849, and 4.216 mg g$^{-1}$ FW), malvidin 3-*O*-glucoside (1.571, 0.650, and 4.564 mg g$^{-1}$ FW), and malvidin 3-*O*-glucoside-5-*O*-glucoside (1.474, 0.452, and 4.358 mg g$^{-1}$ FW). The most abundant compound in the group of flavanols was catechin (0.121, 0.068, and 0.207 mg g$^{-1}$ FW in berry, flesh, and peel, respectively). In the group of flavonols, rutin was most abundant with a mean concentration of 0.062, 0.002, and 0.134 mg g$^{-1}$ FW in berry, flesh, and peel, respectively. Hydroxybenzonic acids, including both gallic and vanillic acids, had relatively low abundance. The mean content of gallic acid was 0.006, 0.003, and 0.011 mg g$^{-1}$ FW in berry, flesh, and peel, respectively, the lowest among all the 48 compounds identified. Caftaric acid (0.461, 0.254, and 0.518 mg g$^{-1}$ FW in berry, flesh, and peel, respectively) was the most abundant compound in the group of hydroxycinnamic derivatives. In general, the phenolic content in red-flesh grape accessions would be higher than *V. vinifera* cultivars and most wild species [29–32].

### 3.2. Total Phenolic Compounds

Table 3 shows the contents of total phenolic compounds, anthocyanins, flavanols, flavonols, hydroxycinnamic derivatives, and hydroxybenzoic acids in the berry, flesh, and peel if the 13 red-fleshed grape accessions. The content of total phenolic compounds in berries ranged from 4.795 to 29.875 mg g$^{-1}$ FW, and the mean was 13.333 mg g$^{-1}$ FW. The accession PI588190 had the highest content of total phenolic compounds, followed by PI588521, PI588103, and PI588579, all of which had more than 15 mg g$^{-1}$ FW in berries. These and other red-fleshed genotypes all had higher contents of total phenolic compounds than non-red-flesh *V. vinifera* and most wild species [29–32].

**Table 3.** The content of total phenolic compounds, anthocyanins, flavanols, flavonols, hydroxycinnamic derivatives, and hydroxybenzoic acids in berry, flesh, and peel of 13 red-fleshed grape accessions (mg g$^{-1}$ FW, Mean $\pm$ SE).

| Accession ID | Tissue | Total Phenolic Compounds | Anthocyanins | Flavanols | Flavonols | Hydroxycinnamic Derivatives | Hydroxybenzoic Acids |
|---|---|---|---|---|---|---|---|
| 588103 | Berry | 17.893 ± 0.093 c * | 17.157 ± 0.120 c | 0.479 ± 0.015 bc | 0.067 ± 0.003 e | 0.177 + 0.009 f | 0.015 ± 0.001 cd |
| 588190 | Berry | 29.875 ± 0.882 a | 28.290 ± 0.739 a | 1.108 ± 0.136 a | 0.221 ± 0.027 b | 0.221 ± 0.027 ef | 0.037 ± 0.009 b |
| 588361 | Berry | 14.571 ± 0.279 d | 13.390 ± 0.240 d | 0.482 ± 0.011 bc | 0.085 ± 0.001 de | 0.606 ± 0.031 c | 0.007 ± 0.000 d |
| 588521 | Berry | 23.377 ± 0.072 b | 19.612 ± 0.047 b | 0.549 ± 0.033 b | 0.431 ± 0.020 a | 2.723 ± 0.073 a | 0.063 ± 0.006 a |
| 588579 | Berry | 15.235 ± 0.426 d | 14.055 ± 0.422 d | 0.437 ± 0.019 bc | 0.134 ± 0.010 cd | 0.596 ± 0.024 c | 0.013 ± 0.001 d |
| 588670 | Berry | 8.882 ± 0.176 f | 7.603 ± 0.115 f | 0.240 ± 0.004 cd | 0.150 ± 0.008 c | 0.874 ± 0.068 b | 0.017 ± 0.002 cd |
| 588681 | Berry | 14.997 ± 0.626 d | 14.297 ± 0.614 d | 0.497 ± 0.013 bc | 0.064 ± 0.003 e | 0.109 ± 0.006 f | 0.030 ± 0.004 bc |
| 597158 | Berry | 6.074 ± 0.033 g | 4.804 ± 0.001 g | 0.162 ± 0.007 d | 0.206 ± 0.005 b | 0.889 ± 0.043 b | 0.013 ± 0.000 d |
| 597175 | Berry | 9.101 ± 0.035 f | 8.021 ± 0.025 f | 0.155 ± 0.003 d | 0.122 ± 0.002 cd | 0.800 ± 0.016 b | 0.004 ± 0.000 d |
| 597193 | Berry | 8.473 ± 0.059 f | 7.591 ± 0.086 f | 0.282 ± 0.010 bcd | 0.143 ± 0.004 c | 0.445 ± 0.014 cd | 0.012 ± 0.001 d |
| 597259 | Berry | 11.868 ± 0.709 e | 10.853 ± 0.697 e | 0.332 ± 0.004 bcd | 0.141 ± 0.010 c | 0.526 ± 0.023 cd | 0.017 ± 0.000 cd |
| 597289 | Berry | 4.795 ± 0.218 g | 3.790 ± 0.217 g | 0.290 ± 0.075 bcd | 0.110 ± 0.001 cde | 0.599 ± 0.048 c | 0.006 ± 0.001 d |
| GVIT1616 | Berry | 8.192 ± 0.480 f | 7.235 ± 0.405 f | 0.509 ± 0.115 bc | 0.060 ± 0.006 e | 0.378 ± 0.042 de | 0.010 ± 0.002 d |
| 588103 | Flesh | 12.593 ± 0.352 a | 12.030 ± 0.356 a | 0.313 ± 0.016 abc | 0.036 ± 0.004 cd | 0.206 ± 0.019 bcd | 0.009 ± 0.001 b |
| 588190 | Flesh | 9.725 ± 0.521 b | 9.266 ± 0.502 b | 0.279 ± 0.008 abcd | 0.030 ± 0.004 cde | 0.130 ± 0.009 d | 0.021 ± 0.002 ab |
| 588361 | Flesh | 8.042 ± 0.550 b | 7.248 ± 0.558 c | 0.342 ± 0.018 ab | 0.086 ± 0.003 a | 0.339 ± 0.006 abcd | 0.027 ± 0.015 ab |
| 588521 | Flesh | 1.960 ± 0.105 d | 1.388 ± 0.090 e | 0.158 ± 0.018 def | 0.019 ± 0.001 ef | 0.387 ± 0.035 abc | 0.009 ± 0.002 b |
| 588579 | Flesh | 8.709 ± 0.776 b | 8.048 ± 0.733 bc | 0.227 ± 0.017 bcdef | 0.061 ± 0.006 b | 0.361 ± 0.029 abcd | 0.012 ± 0.002 b |
| 588670 | Flesh | 3.137 ± 0.234 cd | 2.514 ± 0.202 de | 0.108 ± 0.002 f | 0.023 ± 0.002 def | 0.479 ± 0.030 a | 0.013 ± 0.001 b |
| 588681 | Flesh | 8.015 ± 0.119 b | 7.364 ± 0.019 c | 0.395 ± 0.087 a | 0.058 ± 0.004 b | 0.172 ± 0.015 cd | 0.026 ± 0.000 ab |
| 597158 | Flesh | 2.690 ± 0.007 cd | 2.114 ± 0.017 de | 0.110 ± 0.011 ef | 0.019 ± 0.001 ef | 0.420 ± 0.018 ab | 0.028 ± 0.006 ab |
| 597175 | Flesh | 3.502 ± 0.439 cd | 2.729 ± 0.341 de | 0.177 ± 0.018 cdef | 0.015 ± 0.003 ef | 0.574 ± 0.154 a | 0.005 ± 0.001 b |
| 597193 | Flesh | 4.218 ± 0.186 c | 3.771 ± 0.195 d | 0.192 ± 0.008 cdef | 0.028 ± 0.001 cde | 0.210 ± 0.023 bcd | 0.019 ± 0.005 ab |
| 597259 | Flesh | 3.379 ± 0.430 cd | 2.838 ± 0.418 de | 0.127 ± 0.006 ef | 0.041 ± 0.003 c | 0.358 ± 0.030 abcd | 0.016 ± 0.002 b |
| 597289 | Flesh | 3.941 ± 0.150 c | 3.286 ± 0.184 d | 0.096 ± 0.013 f | 0.010 ± 0.000 f | 0.546 ± 0.022 a | 0.004 ± 0.000 b |
| GVIT1616 | Flesh | 4.321 ± 0.420 c | 3.796 ± 0.401 d | 0.246 ± 0.021 bcde | 0.021 ± 0.002 def | 0.213 ± 0.007 bcd | 0.044 ± 0.004 a |
| 588103 | Peel | 51.018 ± 3.111 abc | 49.412 ± 3.029 abc | 0.905 ± 0.055 bcde | 0.239 ± 0.013 bc | 0.435 ± 0.055 de | 0.030 ± 0.001 b |
| 588190 | Peel | 52.996 ± 3.657 ab | 51.803 ± 3.606 ab | 0.719 ± 0.024 cdef | 0.220 ± 0.026 c | 0.209 ± 0.002 e | 0.045 ± 0.008 b |
| 588361 | Peel | 43.618 ± 3.516 bc | 40.646 ± 3.227 bcd | 1.390 ± 0.130 a | 0.449 ± 0.055 abc | 1.104 ± 0.204 bcd | 0.029 ± 0.005 b |
| 588521 | Peel | 41.124 ± 0.683 bcd | 38.102 ± 0.758 cde | 1.092 ± 0.101 abc | 0.413 ± 0.066 abc | 1.470 ± 0.057 abc | 0.047 ± 0.010 b |
| 588579 | Peel | 59.293 ± 2.954 a | 56.561 ± 2.723 a | 1.304 ± 0.074 ab | 0.540 ± 0.111 a | 0.857 ± 0.065 cde | 0.031 ± 0.005 b |
| 588670 | Peel | 28.100 ± 2.014 ef | 25.970 ± 1.892 fgh | 0.604 ± 0.031 def | 0.218 ± 0.025 c | 1.249 ± 0.145 abcd | 0.058 ± 0.010 b |
| 588681 | Peel | 42.104 ± 2.504 bc | 40.252 ± 2.371 cd | 1.137 ± 0.088 abc | 0.460 ± 0.067 abc | 0.202 ± 0.007 e | 0.052 ± 0.006 b |
| 597158 | Peel | 17.067 ± 0.377 f | 15.138 ± 2.371 h | 0.492 ± 0.015 ef | 0.428 ± 0.023 abc | 0.992 ± 0.082 bcde | 0.017 ± 0.003 b |
| 597175 | Peel | 25.213 ± 2.363 f | 22.426 ± 2.136 gh | 0.438 ± 0.039 f | 0.258 ± 0.040 bc | 2.080 ± 0.234 a | 0.013 ± 0.002 b |
| 597193 | Peel | 21.269 ± 1.060 f | 19.468 ± 0.875 gh | 0.513 ± 0.073 ef | 0.398 ± 0.065 abc | 0.837 ± 0.100 cde | 0.052 ± 0.004 b |
| 597259 | Peel | 39.093 ± 2.258 cde | 35.337 ± 1.694 def | 1.405 ± 0.153 a | 0.497 ± 0.051 ab | 1.800 ± 0.430 ab | 0.055 ± 0.016 b |
| 597289 | Peel | 60.182 ± 3.097 a | 57.515 ± 3.081 a | 0.958 ± 0.013 bcd | 0.450 ± 0.039 abc | 1.122 ± 0.086 bcd | 0.135 ± 0.050 a |
| GVIT1616 | Peel | 29.343 ± 1.657 def | 27.382 ± 1.385 efg | 0.630 ± 0.127 def | 0.220 ± 0.030 c | 1.101 ± 0.195 bcd | 0.011 ± 0.001 b |

*: Those with different letters in the same rows are significantly different at $p < 0.05$.

The content of total phenolic compounds in flesh ranged from 1.960 to 12.593 mg g$^{-1}$ FW, and the mean was 5.710 mg g$^{-1}$ FW. PI588103 had the highest content of total phenolic compounds, followed by PI588190, PI588579, PI599361, and PI588681, all of which had more than 8 mg g$^{-1}$ FW (Table 3).

The content of total phenolic compounds in peel was significantly higher than that in berry and flesh ($p < 0.05$), ranging from 17.067 to 60.182 mg g$^{-1}$ FW with a mean value of 39.263 mg g$^{-1}$ FW (Table 4). The accession PI597289 had the highest content of total phenolic compounds, followed by PI588579, PI588190, and PI588103, all of which had more than 50 mg g$^{-1}$ FW. As expected, peel has the most abundant phenolic compounds in grape berry, and contents were typically seven-fold and three-fold higher than in flesh and whole berry, respectively. Therefore, peel is the main determinator of the total phenolic compounds in grape berries.

### 3.3. Anthocyanins

Anthocyanins were the main polyphenolic compounds in red-flesh grape accessions, ranging from 3.790 to 28.290 mg g$^{-1}$ FW, 1.388 to 12.030 mg g$^{-1}$ FW, and 15.138 to 57.515 mg g$^{-1}$ FW in berry, flesh, and peel, respectively (Table 3). The content of anthocyanins in peel was significantly higher than that in whole berry and flesh (Table 4). On average, anthocyanins accounted for 90.4, 89.4, and 94.1% of the total phenolic compounds in berry, flesh, and peel, respectively (Table 2, Figure 1). The relative contents were higher than that observed for non-red-flesh *V. vinifera* and most wild species [29–34]. PI588190,

PI588103, and PI599289 had the highest mean anthocyanin contents in berry, flesh, and peel (Table 3), respectively, similar to those found for the total phenolic compounds. In contrast, PI597289, PI588521, and PI597158 had the lowest mean contents of anthocyanins, but they were still higher than that of non-red-flesh *V. vinifera* [29].

**Table 4.** Mean comparisons of the content of phenolic compounds among whole berry, flesh, and peel in 13 red-fleshed grape accessions (mg g$^{-1}$ FW, Mean $\pm$ SE) *.

| | Berry | Flesh | Peel |
|---|---|---|---|
| Total phenolic compounds | 13.333 $\pm$ 1.992 b * | 5.710 $\pm$ 0.915 c | 39.263 $\pm$ 3.949 a |
| Anthocyanins | 12.054 $\pm$ 1.893 b | 5.107 $\pm$ 0.915 b | 36.924 $\pm$ 3.936 a |
| Flavanols | 0.425 $\pm$ 0.068 b | 0.213 $\pm$ 0.027 c | 0.891 $\pm$ 0.097 a |
| Flavonols | 0.149 $\pm$ 0.027 b | 0.034 $\pm$ 0.006 c | 0.368 $\pm$ 0.033 a |
| Hydroxycinnamic derivatives | 0.688 $\pm$ 0.183 ab | 0.338 $\pm$ 0.040 b | 1.035 $\pm$ 0.155 a |
| Hydroxybenzoic acids | 0.019 $\pm$ 0.004 b | 0.018 $\pm$ 0.003 b | 0.044 $\pm$ 0.009 a |

*: Those with different letters in the same rows are significantly different at *p* < 0.05.

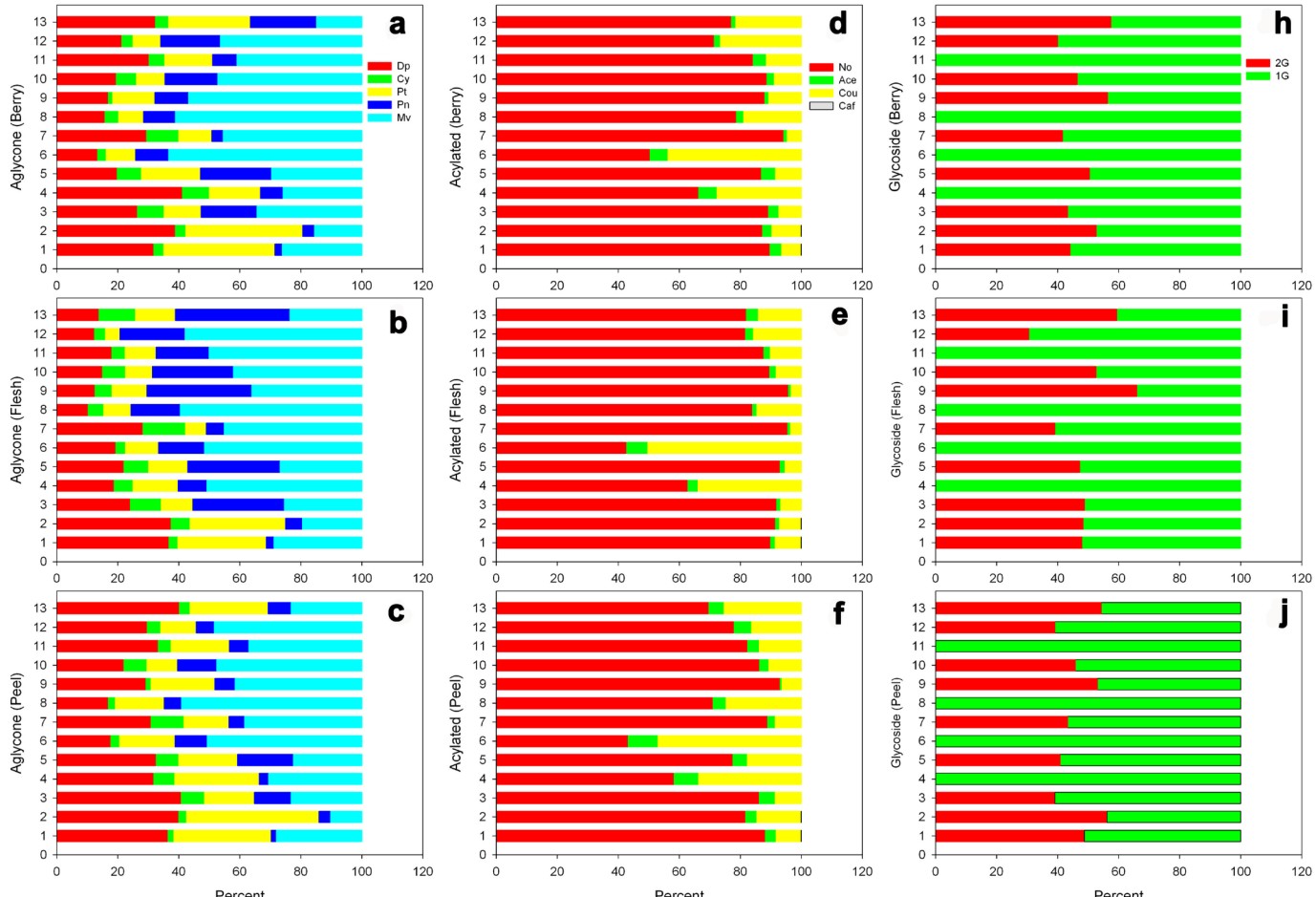

**Figure 1.** Relative percent content of aglycone, acylated, and glycoside anthocyanins in berry, flesh, and peel of 13 red-flesh grape accessions: aglycone anthocyanins in berry (**a**), flesh (**b**), and peel (**c**); acylated anthocyanins in berry (**d**), flesh (**e**), and peel (**f**); and glycoside anthocyanins in berry (**h**), flesh (**i**), and peel (**j**). Cy = cyanidin; Dp = delphinidin; Pt = petunidin; Pn = peonidin; Mv = malvidin, with all their corresponding derivatives included. No = non acylated anthocyanins; Ace = acetylated anthocyanins; Cou = coumarylated anthocyanins; Caf = caffeoylated anthocyanins; 2G = diglucoside anthocyanins; 1G = monoglucoside anthocyanins. The identities of grape accessions follow those in Table 1.

A total of 28 different anthocyanin compounds were detected in this study (Table 2), and the relative abundance of these anthocyanins is shown in Figure 1. Mv derivatives were the most abundant, accounting for 32.4, 26.4, and 33.2% of the total anthocyanins in berry, flesh, and peel, respectively, followed by Dp derivatives (29.6, 25.4, and 32.7%, respectively), Pt (22.5, 24.3, and 21.9%, respectively) and Pn derivatives (9.7, 16.8, and 7.1%, respectively). Cy derivatives were the least abundant, accounting for 5.8, 7.1, and 5.1% of the total anthocyanins in berry, flesh, and peel, respectively (Figure 1a–c). Mv derivatives had a wide range of variation among accessions, ranging from 14.9 (GVIT1616) to 63.4% (PI588670) of the total anthocyanins in berry, from 19.5 (PI588190) to 59.6% (PI 597158) of the total anthocyanins in flesh, and from 10.3 (PI588190) to 59.1% (PI588158) of the total anthocyanins in peel. For most accessions, the relative abundance of Mv derivatives in flesh and peel were similar except for PI588521 and PI597259 (the relative abundance of Mv derivatives in peel was higher than that in flesh). Different from Mv derivatives, the relative abundance of Dp and Pt derivatives was lower in flesh than that in peel for most accessions. PI588521 had the highest relative content of Dp derivatives, accounting for 41.1% of the total anthocyanins, followed by PI588190 (38.9%), GVIT1616 (32.3%), PI588103 (31.8%), and PI597251 (30.2%) in berry. PI588190 and PI588103 had the highest relative content of Dp derivative in flesh, respectively accounting for 37.5 and 36.7% of the total anthocyanins. PI588361 and GVIT1616 had the highest relative content of Dp derivative in peel, respectively accounting for 40.8 and 40.2% of the total anthocyanins. PI588190 had the highest relative content of Pt derivatives, accounting for 38.3, 31.3, and 43.4% of the total anthocyanins in berry, flesh, and peel, respectively. In contrast, the relative content of Pn and Cy derivatives in flesh was higher than that in peel in most accessions, with PI588579, PI588175, and GVIT1616 having the highest relative content of Pn-derivatives. Furthermore, PI588681 had the highest relative content of Cy derivatives (10.6, 13.9, and 10.9% of the total anthocyanins in berry, flesh, and peel, respectively).

Anthocyanins can also be classified on the basis of whether they are acylated or not. Nonacylated anthocyanins were the most abundant anthocyanins in the 13 red-flesh accessions (Figure 1), accounting for 81.0, 83.7, and 77.3% of the total anthocyanins in berry, flesh, and peel, respectively. PI588681 had the highest relative content of nonacylated anthocyanins (94.3% of the total anthocyanins) in berry, and PI 597175 had the highest relative content in flesh and peel (accounting for 95.8 and 93.0% the total anthocyanins in flesh and peel, respectively), while PI588670 had the lowest relative content of nonacylated anthocyanins in berry, flesh, and peel (accounting for 50.4, 42.8, and 43.4% of the total anthocyanins, respectively). Coumaryl anthocyanins were the second most abundant anthocyanins, accounting for 15.8, 14.0, and 18.1% of the total anthocyanins in berry, flesh, and peel, respectively. Acetyl and caffeoyl anthocyanins were generally low in all accessions, which accounted for no more than 5% of the total anthocyanins in berry, flesh, or peel. In comparison, non-red-flesh *V. vinifera* had similar profiles of compositions, but the proportion of nonacylated anthocyanins appeared to be generally lower [29–34].

Anthocyanins in grapes are all glycosides (Figure 1). In PI588670, PI588521, PI597259, and PI597158, only monoglucoside anthocyanins were detected. In the other nine accessions with a non-*vinifera* germplasm background, both monoglucoside and diglucoside derivatives were found. Monoglucoside derivatives, on average, accounted for 48.2, 49.1, and 46.8% of the total anthocyanins in berry, flesh, and peel, respectively.

### 3.4. Flavanols and Flavonols

The contents of total flavanols in 13 grape accessions are shown in Table 3. The content of total flavanols ranged from 0.155 to 1.108 mg g$^{-1}$ FW in berry, from 0.096 to 0.395 mg g$^{-1}$ FW in flesh, and from 0.438 to 1.405 mg g$^{-1}$ FW in peel, with a mean value of 0.325, 0.213, and 0.891 mg g$^{-1}$ FW in berry, flesh, and peel, respectively. The peel had the highest content of flavanols, followed by flesh and whole berry. The differences in the contents of these three types of tissue were all significant (Table 4).

Flavanols comprised procyanidin B1, B2, catechin, epicatechin, epicatechin gallate, and isorhamnetin. Catechin was the most abundant flavanol in berry and flesh, accounting for 28.5 and 31.9% of the total flavanols, respectively (Figure 2). Procyanidin B1 was the second most abundant flavanol, accounting for 26.1 and 26.5% of the total flavanols in berry and flesh, respectively. In peel, however, procyanidin B1 was the most abundant flavanol, accounting for 34.5% of the total flavanols, while catechin was the second most abundant flavanol, accounting for 23.2% of the total flavanols. Isorhamnetin was the least abundant, accounting for only about 3.8, 4.7, and 9.0% of the total flavanols in berry, flesh, and peel, respectively.

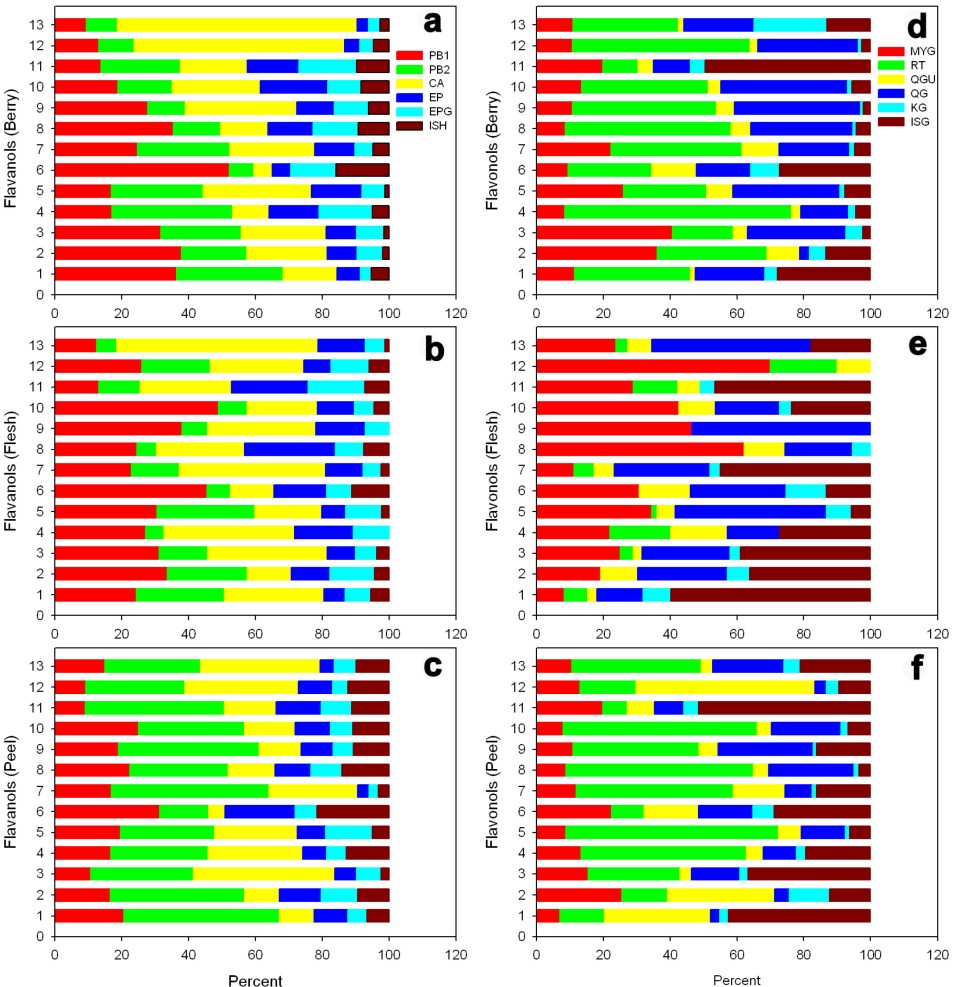

**Figure 2.** Relative percent content of flavanols and flavonols in berry, flesh, and peel of 13 red-flesh grape accessions: flavanols in berry (**a**), flesh (**b**), and peel (**c**), and flavonols in berry (**d**), flesh (**e**), and peel (**f**). PB1 = procyanidin B1; PB2 = procyanidin B2; CA = catechin; EP = epicate-chin; EPG = epicatechin gallate; ISH = isorhamnetin; MYG = myricetin 3-*O*-glucoside; RT = rutin; QG = quercetin 3-*O*-glucoside; QGU = quercetin 3-O-gluruconide; KG = kaempferol 3-*O*-glucoside; ISG = isorhamnetin 3-O-glucoside. The identities of grape accessions follow those in Table 1.

The mean content of flavanols in the whole berries of red-flesh grape accessions was about three-fold higher than that in non-red-flesh *V. vinifera* cultivars [29–32,46]. The relative content of procyanidin B1 in the peel of red-flesh grapes was similar to wild grape species, but lower than that in *V. vinifera* [29]. The relative content of other flavanols in red-flesh grapes was similar to *V. vinifera* cultivars and wild grape species [29–32].

The contents of the total flavonols in individual accessions are also shown in Table 3. The content of total flavonols ranged from 0.060 to 0.431 mg g$^{-1}$ FW in berry, from 0.010 to 0.086 mg g$^{-1}$ FW in flesh, and from 0.218 to 0.540 mg g$^{-1}$ FW in peel, with a mean value

of 0.149, 0.034, and 0.368 mg g$^{-1}$ FW in berry, flesh, and peel, respectively (Table 3). The content of flavonols in red-flesh grapes was about three-fold higher than that in non-red-flesh *V. vinifera* cultivars and was also higher than that in wild species [29–32,46].

Flavonols accounted for about 1.1, 0.6, and 0.9% of the total phenolic compounds and 11.6, 5.7, and 15.8% of the non-anthocyanin phenolic compounds in berry, flesh, and peel, respectively (Tables 2 and 3). The peel had the highest content of flavonols, followed by flesh and whole berry, and their content differences were all significant (Table 4). PI588521, PI588361, and PI588579 had the highest content of flavonols in berry, flesh, and peel (0.431, 0.086, and 0.540 mg g$^{-1}$ FW, respectively). Flavonols detected in the present study included rutin, myricetin 3-*O*-glucoside, quercetin 3-*O*-glucoside, quercetin 3-*O*-glucuronide, kaempferol 3-*O*-rutinoside, and isorhamnetin 3-*O*-glucoside. Rutin was most abundant in berry and peel, and on average accounted for 41.6 and 36.4% of the total flavonols, respectively. Isorhamnetin 3-*O*-glucoside was most abundant in flesh, which on average accounted for 29.4% of total flavonols (Figure 2). In comparison, quercetin 3-*O*-glucuronide was the most abundant compound and accounted for about 40% of the total flavonols in *V. vinifera*, and myricetin 3-*O*-glucoside was the most abundant flavonol in wild grape species [29].

### 3.5. Hydroxycinnamic Derivatives and Hydroxybenzoic Acids

The range of variation in the total content of hydroxycinnamic derivatives in the red-flesh grapes is presented in Table 3. The content of hydroxycinnamic derivatives ranged from 0.177 to 2.723 mg g$^{-1}$ FW in berry, from 0.130 to 0.574 mg g$^{-1}$ FW in flesh, and from 0.202 to 2.080 mg g$^{-1}$ FW in peel, with an average of 53.8, 56.1, and 44.3% accounting for non-anthocyanin phenolic compounds in berry, flesh, and peel, respectively. The content of hydroxycinnamic derivatives in peel was significantly higher than that in berry and flesh (Table 4). Hydroxycinnamic derivatives comprised caftaric acid, chlorogenic acid, coutaric acid, feruic acid, tryptophol, and resveratrol. Caftaric acid was the most abundant hydroxycinnamic derivative in berry, flesh, and peel with a mean value of 0.461, 0.254, and 0.518 mg g$^{-1}$ FW, respectively, which accounted for 63.2, 71.9, and 45.6% of the total hydroxycinnamic derivatives in the three tissues, respectively (Table 2). PI588670, PI597289, and PI597158, respectively, had the highest content of caftaric acid in berry, flesh, and peel (Figure 3). Coutaric acid was the second most abundant hydroxycinnamic derivatives in berry, flesh, and peel, accounting for 24.6, 16.5, and 38.8% of the total hydroxycinnamic derivatives, respectively. Resveratrol, an important phytochemical with many well-known benefits for human health, was 0.003, 0.001, and 0.013 mg g$^{-1}$ FW, accounting for 0.9, 0.4, and 2.2% of the total hydroxycinnamic derivatives in berry, flesh, and peel, respectively (Figure 3). PI588190, PI588361, and PI588579 had the highest content of resveratrol in berry, flesh, and peel, with a value of 0.009, 0.003, and 0.025 mg g$^{-1}$ FW, respectively. The content of resveratrol in red-flesh grapes was higher than that in *V. vinifera*, but similar to many wild species [29–34].

Hydroxybenzoic acids had the lowest content in the five polyphenolic groups, accounting for 1.5, 3.0, and 1.9% of non-anthocyanin phenolic compounds in berry, flesh, and peel, respectively (Table 3). The content of hydroxybenzoic acids in peel was significantly higher than that in flesh and whole berry (Table 4). The total content of hydroxybenzoic acids ranged from 0.006 to 0.063 mg g$^{-1}$ FW in berry, 0.004 to 0.044 mg g$^{-1}$ FW in flesh, and 0.011 to 0.135 mg g$^{-1}$ FW in peel. PI588521, GVIT1616, and PI597289 had the highest content of hydroxybenzoic acids in berry, flesh, and peel, respectively (Figure 3). Two hydroxybenzoic compounds, gallic acid and vanillic acid, were detected. Gallic acid, on average, accounted for 30.5, 22.1, and 32.5% of the total hydroxybenzoic acids in berry, flesh, and peel, respectively. The content of vanillic acid was much higher than that of gallic acid in berry, flesh, and peel.

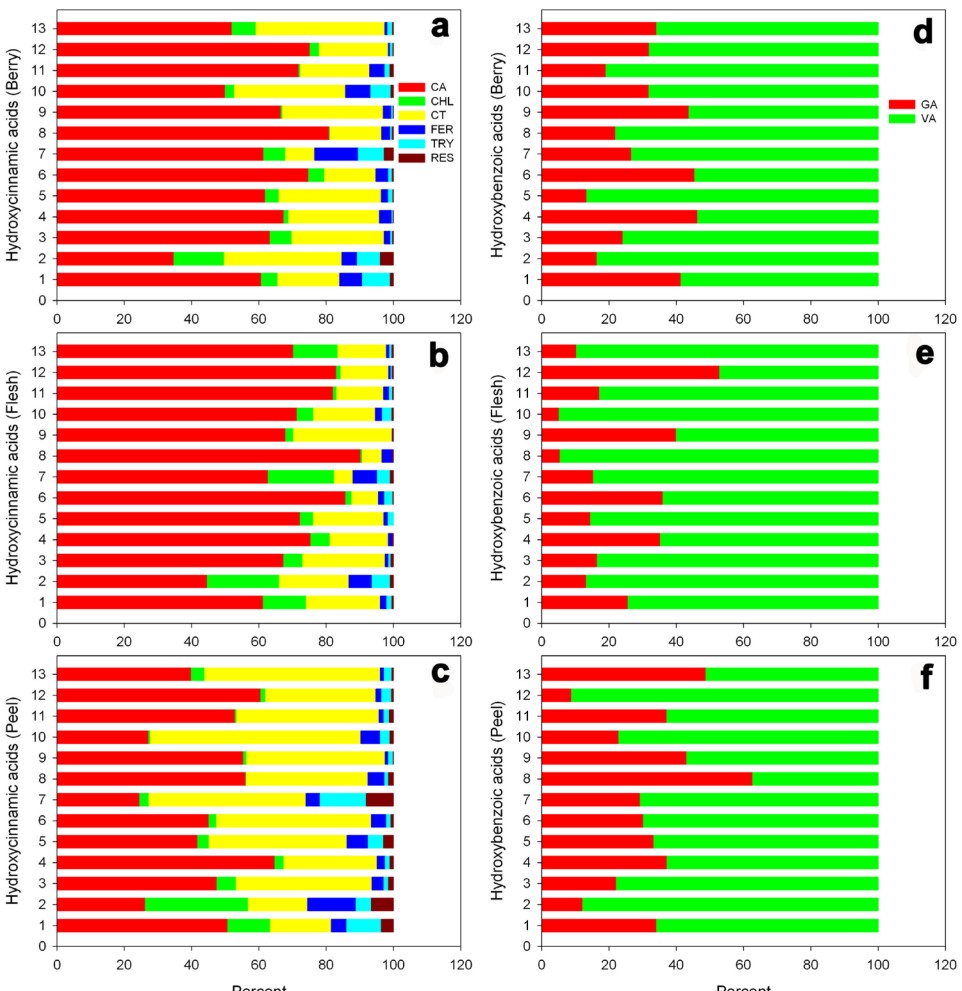

**Figure 3.** Relative percent content of phenolic acids in berry, flesh, and peel of 13 red-flesh grape accessions: hydroxycinnamic derivatives in berry (**a**), flesh (**b**), and peel (**c**), and hydroxybenzoic acids in berry (**d**), flesh (**e**), and peel (**f**). CA = caftaric acid; CHL = chlorogenic acid; CT = coutaric acid; FER = ferulic acid; TRY = tryptophol RES = resveratrol; GA = gallic acid; VA = vanillic acid. The identities of grape accessions follow those in Table 1.

## 4. Conclusions

Grapes are a rich source of polyphenolic compounds. As expected, the red-fleshed grapes, compared to most *V. vinifera* cultivars and wild species, contained much higher contents of total phenolic compounds, anthocyanins in particular. As compared with flesh and whole berry, the peel had pronounced higher phenolic compounds, the content of which was the most abundant. This study fully provided a comprehensive assessment of the variation patterns of 48 phenolic compounds in 13 red-fleshed grape accessions. Such a broad spectrum of study on such a number red-fleshed grapes accessions, counterparts, and analyzed compounds has not been reported before, and a cascade of information about non-anthocyanin compounds was obtained for the first time. Furthermore, tremendous variation emerged among the detected compounds in red-fleshed grape accessions, and several red-flesh grape accessions that were rich in high contents of total and individual phenolic compounds were also identified. The results have important value for the future application of these red-fleshed grape accessions in industry or as parents for red-fleshed-grape breeding purposes.

**Author Contributions:** Conceptualization, G.-Y.Z., Z.L. and L.C.; methodology, Z.L.; software, L.L. and Z.L.; validation, L.L. and Z.L.; formal analysis, L.L. and Z.L.; investigation, L.L.; resources, Y.Y.; data curation, Z.L.; writing—original draft preparation, L.L. and Z.L.; writing—review and editing, G.-Y.Z. and L.C.; visualization, L.L.; supervision, Z.L.; project administration, Z.L.; funding acquisition, G.-Y.Z., Z.L. and L.C. All authors have read and agreed to the published version of the manuscript.

**Funding:** This research received no external funding.

**Data Availability Statement:** Not applicable.

**Acknowledgments:** We are grateful to Dawn Dellefave, Heidi Schwaninger and Bill Srmack of the USDA-ARS Plant Genetic Resources Unit in Geneva, New York for their assistance in collecting berry samples.

**Conflicts of Interest:** The authors declare no conflict of interest.

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
