# Peer review of "Phytochemical Composition and Content of Red-Fleshed Grape Accessions"

_horticulturae, doi:10.3390/horticulturae9050579_

Round 1

Reviewer 1 Report (Previous Reviewer 2)

The manuscript improved considerably; however, I see it as necessary to address 3 comments to improve its quality:

1.     Tables must appear immediately after their mention in the main text. Table 1 should be placed after line 98.

2.     It is necessary to express the level of significance (p<0.05) in the methodology.

3.     There should be no citations in the conclusions (there was already a whole section to compare results with other studies).

Author Response

  1. Tables must appear immediately after their mention in the main text. Table 1 should be placed after line 98.

Response: We figured out the position of Table 1 in the text.

  1. It is necessary to express the level of significance (p<0.05) in the methodology.

Response: We added it in Statistical analyses Section.

  1.     There should be no citations in the conclusions (there was already a whole section to compare results with other studies).

Response: We canceled it in the text.

Reviewer 2 Report (New Reviewer)

I have some suggestions how to improve your manuscript.

Table 1. Please insert Table 1 in section material and methods chapter 2.1 plant material not in section results and discussion.

Chapter 2.4 Statistical analyses. In Tables 2,3 and 4 you used statistical significant differences but in chapter 2.4 you did not mention what method you used for this analysis. Please write what method did you use in chapter 2.4.

Table 2. You mention a lot of phenolic compounds. Please write the main phenolic compounds in bold not in simple font, it will be more clear for readers.

Conclusions. In conclusions you compared your study with previous other studies. I think you have to compare your study with other previous studies in section results and discussion not in section conclusions. In section conclusions you have to write the main statements of your study.

Literature source 16.  I think year 2004 have to be write in bold font not in simple. Please check.

Author Response

  1. Table 1. Please insert Table 1 in section material and methods chapter 2.1 plant material not in section results and discussion.

Response: We revised it in the text.

  1. Chapter 2.4 . In Tables 2,3 and 4 you used statistical significant differences but in chapter 2.4 you did not mention what method you used for this analysis. Please write what method did you use in chapter 2.4.

Response: In Statistical analyses Section, we introduced that SAS programmes of GLM were used for mean separation and significance testing (p<0.05).

  1. Table 2. You mention a lot of phenolic compounds. Please write the main phenolic compounds in bold not in simple font, it will be more clear for readers.

Response: We revised it in Table 2.

  1. In conclusions you compared your study with previous other studies. I think you have to compare your study with other previous studies in section results and discussion not in section conclusions. In section conclusions you have to write the main statements of your study.

Response: We revised it in the text.

  1. Literature source 16.  I think year 2004 have to be write in bold font not in simple. Please check.

Response: We revised it in the text.

Reviewer 3 Report (New Reviewer)

The authors have made minor errors, which you kindly address:   

1. line 58 the unit is written in the form mg/kg, while in the rest of the text the writing is in the form e.g. mg kg-1. please unify the writing of the units.

2. line 109 should be written after 2:28:70, v/v/v

3. line 124 should be written after (90:10, v/v)

Author Response

1. line 58 the unit is written in the form mg/kg, while in the rest of the text the writing is in the form e.g. mg kg-1. please unify the writing of the units.

Response: We revised it in the text.

2.line 109 should be written after 2:28:70, v/v/v,

Response: We revised it in the text.

3.line 124 should be written after (90:10, v/v)

Response: We revised it in the text.

This manuscript is a resubmission of an earlier submission. The following is a list of the peer review reports and author responses from that submission.

Round 1

Reviewer 1 Report

In this study authors aimed to characterize the phenolic composition of whole berry, flesh and peel (skin) of several red-flashed grape accessions. Although the performed analyses are very thorough, the manuscript has several flaws and it is not suitable for publication in Horticulturae journal.

The manuscript does not bring any novelty from the scientific point of view, and it only very basically characterizes the phytochemical composition of the investigated accessions. The differences in phenolic composition among different accessions are very large, but this is normal for most grape accessions, and also for other chemical compounds, and as such does not bring scientifically anything new. Some other flaws are observed, like the use of the term ‘polyphenolic’ for the compounds which are considered as simple (or mono-) phenolic compounds (not poly, because they have only one benzene ring), like hydroxycinnamic derivatives or hydroxybenzoic acids, so the proper term would be ‘phenolic compounds’. And concerning the terminology, why the term ‘peel’ is used, and not ‘skin’? Also, the results of the % of different classes of phenolic compounds are suspicious. For example, in the whole berry usually the flavanols are the most abundant because of their high content in the seeds (and not the anthocyanins). Also, if taking in the account only the non-anthocyanin phenolic compounds, flavanols are usually more abundant than hydroxycinnamic derivatives (and in this study it was the opposite). Because of all these issues, I recommend publishing these results in some technical journal concerning agriculture, or the field of viticulture and enology.

Reviewer 2 Report

General comment:

It is a well-structured manuscript. However, the way to present some tables is confusing. It is important to be very clear in the methodologies, in the titles of the Tables and Figures. On the other hand, it is understood that the study is totally focused on COMPARING the phytochemical content of 13 accessions, so a statistical analysis that shows and demonstrates the differences between them is INDISPENSABLE. The study lacks the specifications of such an analysis in the methodology and in the results are briefly mentioned. I believe that the statistical analysis of all its results (Tables 2-4 and Figures 1-3) would increase the quality of the study.

Specific comments:

Title

The title is adequate.

Abstract

It is necessary to start with a couple of introductory sentences that contextualize the reason or benefit of the study.

Keywords

It is necessary to give them an order that relates them, for example: Red-flesh grape; Polyphenols; Anthocyanins; Flavanols; Teinturier.

1. Introduction

The introduction is short but with enough information to contextualize the reader. It is appropriate to include the chemical structures of anthocyanins, hydroxybenzoic acids, hydroxycinnamic derivatives, flavonols and flavanols (who were classified in the study).

2. Materials and Methods

Some observations must be attended:

·       Lines 86-96. Tables must appear immediately after their mention in the main text. Table 1 should be placed after line 96.

·       Lines 97-110. It is not clear if the extracts of the 13 red flesh Vitis accessions obtained were mixed or analyzed individually.

·       Lines 98-99. The correct citation of a text is in the way they had it before its correction (the citation after the mention of the authors): Liang et al., [29]. For several years now, the term “et al.,” should no longer be italicized.

·       Lines 112-113. The correct citation of a text is in the way they had it before its correction (the citation after the mention of the authors): Liang et al., [29]. For several years now, the term “et al.,” should no longer be italicized.

·       Line 136. For several years now, the term “et al.,” should no longer be italicized.

·       Lines 139-143. It is necessary to specify what type of analysis of variance and method of comparison of means were carried out.

3. Results and discussion

The presentation of the results is confusing because some tables show the results of the 13 accessions studied and others show unique data that does not indicate what it corresponds to; therefore, the assertions made by the authors cannot be verified. Some observations must be attended:

·       Lines 146-154. The text indicates that the results are from 13 red flesh Vitis accessions, however, general results are shown in Table 2.

·       Lines 168-171. The authors assert that their results were "greater" compared to the wild species, however, since they did not specify an analysis of means, this assertion can only be in numerical and non-statistical terms.

·       Lines 191-197. The authors include the level of significance (p<0.05 (the correct thing is to express it with lower case)) but it is not reported in the methodology. The main text mentions specific accessions; however, Table 4 is very general and does not specify which accession it corresponds to. It is necessary to report the complete statistical analysis on the methodology and the accession specification in Table 4, because the statement “was significantly higher” cannot be verified.

·       Table 3. Why were these results not statistically evaluated?

·       Table 4. It is necessary to specify which accession the results are from or to whom "berry", "flesh" and "peel" belong.

4. Conclusions

Some comments:

·       The conclusions are not cited (there was already a whole section to compare results with other studies).

·       The authors minimized their work by including the phrase “While these general conclusions were not novel,” it is necessary to highlight the intention and the information that the study provides.

·       It is important to mention the most outstanding results of the study and its possible use or application.

5. References

The number of references is adequate, however, the studies could be more current (last 5 years), since most of them exceed the decade of their publication.